# Diversity and Activity of Aquatic Cellulolytic Bacteria Isolated from Sedimentary Water in the Littoral Zone of Tonle Sap Lake, Cambodia

Aiya Chantarasiri 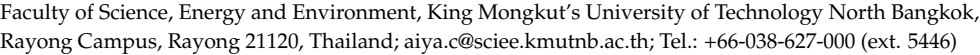

Faculty of Science, Energy and Environment, King Mongkut's University of Technology North Bangkok, Rayong Campus, Rayong 21120, Thailand; aiya.c@sciee.kmutnb.ac.th; Tel.: +66-038-627-000 (ext. 5446)

**Abstract:** Tonle Sap Lake is the largest freshwater lake in Southeast Asia, and it is regarded as one of the most biodiverse freshwater ecosystems in the world. Studies concerning aquatic cellulolytic bacteria from Tonle Sap Lake remain scarce. Cellulolytic bacteria and their cellulases play a vital role in the biogeochemical cycles of lake environments, and their application in biotechnological industries is likewise an important component of their usage. This study aimed to assess the isolation, genetic identification, bioinformatic analyses, and activity characterization of aquatic cellulolytic bacteria. The cellulolytic bacteria isolated from sedimentary water samples in the littoral zone of the lake belong to the genera *Aeromonas*, *Bacillus*, and *Exiguobacterium*. Several isolated aquatic bacteria were designated as rare cellulolytic microbes. Remarkably, *B. mojavensis* strain REP303 was initially evidenced by the aquatic cellulolytic bacterium in freshwater lake ecosystems. It was considered a highly active cellulolytic bacterium capable of creating a complete cellulase system involving endoglucanase, exoglucanase, and β-glucosidase. The encoded endoglucanase belongs to the glycosyl hydrolase family 5 (GH5), with a carboxymethylcellulase (CMCase) activity of $3.97 \pm 0.05$ U/mL. The optimum temperature and pH for CMCase activity were determined to be 50 °C at a pH of 7.0, with a stability range of 25–55 °C at a neutral pH of 7.0–8.0. The CMCase activity was enhanced significantly by $Mn^{2+}$ and was inhibited considerably by EDTA and ethyl-acetate. In conclusion, this study is the first to report data concerning aquatic cellulolytic bacteria isolated from the littoral zone of Tonle Sap Lake. A novel strain of isolated cellulolytic *B. mojavensis* could be applied in various cellulose-based industries.

**Keywords:** *Aeromonas*; *Bacillus*; *Bacillus mojavensis*; cellulase; *Exiguobacterium*; sedimentary water; Tonle Sap Lake





## 1. Introduction

Freshwater habitats are a tremendous source of organic matter and play a vital role in global biogeochemical cycles [1]. A lake is a type of freshwater habitat. It is a large basin surrounded by land that is fed by rivers and unsalted runoff. Each lake has unique characteristics and morphological properties that distinguish it from other lakes [2]. Microorganisms play a key role in the decomposition of organic matter and nutrient regeneration, and are important members of microbial food webs, as well as sensitive indicators of water quality [3–5]. Bacteria are widespread in lake environments and are primarily involved in various biogeochemical cycles [6]. Bacterial community composition among lakes can be quite diverse [7] and varies with different environmental variables [8,9]. Bacteria dwelling in freshwater lakes commonly belong to the phyla Actinobacteria, Bacteroidetes, Firmicutes, Proteobacteria, and Verrucomicrobia [7,10,11]. Studies concerning bacterial diversity in freshwater lakes are critically important for various reasons, such as acquired knowledge about genetic resources, distribution patterns, and functional annotations [6]. In addition, many aquatic bacteria are sources of hydrolytic enzymes and other important industrial enzymes [6,12].

Cellulolytic bacteria have been found in a wide range of habitats and environments, such as animal digestive tracts [13,14], decaying organic matter [15–17], herbivore dung [18,19], mangrove sediments [20,21], manure [22,23], terrestrial soils [24,25], and wetland soils [26,27]. However, few studies have focused on the cellulolytic microbes isolated from lake environments [28,29]. Recently, the cellulolytic bacteria isolated from freshwater lakes were identified as belonging to the genera *Aneurinibacillus*, *Bacillus*, *Klebsiella*, *Micromonospora*, *Proteus*, *Pseudomonas*, and *Streptomyces* [29–31]. These cellulolytic bacteria are mainly related to the carbon cycle of organic matter in freshwater lake ecosystems and are responsible for the hydrolysis of lignocellulosic biomass to fermentable sugars by their cellulolytic enzymes [27,32]. Cellulolytic enzymes, generally called cellulases, comprise endoglucanases or carboxymethylcellulases (E.C. 3.2.1.4), exoglucanases or cellobiohydrolases (E.C. 3.2.1.91), and β-glucosidases (E.C. 3.2.1.21), which synergistically work to hydrolyze the β-1,4 glycosidic linkages of cellulose polymer in lignocellulosic biomass [27,33,34]. Nowadays, cellulases account for 20% of the global enzyme market and they have biotechnological potential in various industries [35,36]. Therefore, the isolation and screening of cellulolytic microbes from various environments are some of the important approaches for obtaining novel cellulases [35]. Most bacteria in nature cannot be isolated and cultivated by traditional culture-based methods. Molecular and in silico analyses of unculturable bacteria have provided more information on phylogenetic and genomic studies than the traditional techniques [37]. However, the majority of cellulases have been isolated from cultured microorganisms [38]. Culturable bacteria are still considered because they are practical for various biotechnological industries and small community enterprises. Moreover, culture-based methods of cultural bacteria remain necessary to validate phenotype predictions made on the basis of genomic analyses [39].

To improve knowledge about the cellulolytic bacteria dwelling in freshwater lakes and their cellulolytic activity, more studies should be conducted. Therefore, this study aimed to isolate, screen, and cultivate aquatic cellulolytic bacteria from a complex and unique freshwater ecosystem, namely Tonle Sap Lake. It is the largest freshwater lake in Southeast Asia and is located in the central part of Cambodia [40,41]. This lake is known as one of the most biodiverse and productive freshwater ecosystems in the world [40,42]. However, information concerning the aquatic cellulolytic bacteria isolated from this lake remains extremely limited. Molecular genetic methods and nucleotide sequencing analysis of the 16S rRNA gene were used to identify and describe the diversity of isolated cellulolytic bacteria. All aquatic cellulolytic bacteria were determined for their cellulolytic activities, including endoglucanases, exoglucanases, β-glucosidases, and total cellulases. Finally, the endoglucanase produced from the most effective bacterium, *Bacillus mojavensis* strain REP303, was bioinformatically analyzed and enzymatically characterized to evaluate its cellulolytic potential.

## 2. Materials and Methods

### 2.1. Study Area

The study area was Tonle Sap Lake (the Great Lake), which is located in the floodplain region of the central basin of Cambodia. This lake covers an area of 2500 km$^2$ during the dry season (December to April) and expands up to 16,000 km$^2$ during the rainy season (May to October) [40,41]. The ecological topography consists of freshwater areas, flooded forests, marshes, rice fields, shrubs, wetlands, and rural areas. The freshwater in the lake exchanges periodically with the Mekong River, which affects the water quality in the lake from season to season [41]. The sampling points were in the Northwestern part of the lake near road No.63 of Siem Reap Province (13°23′ N, 103°82′ E). The location of the study area is shown in Figure 1.

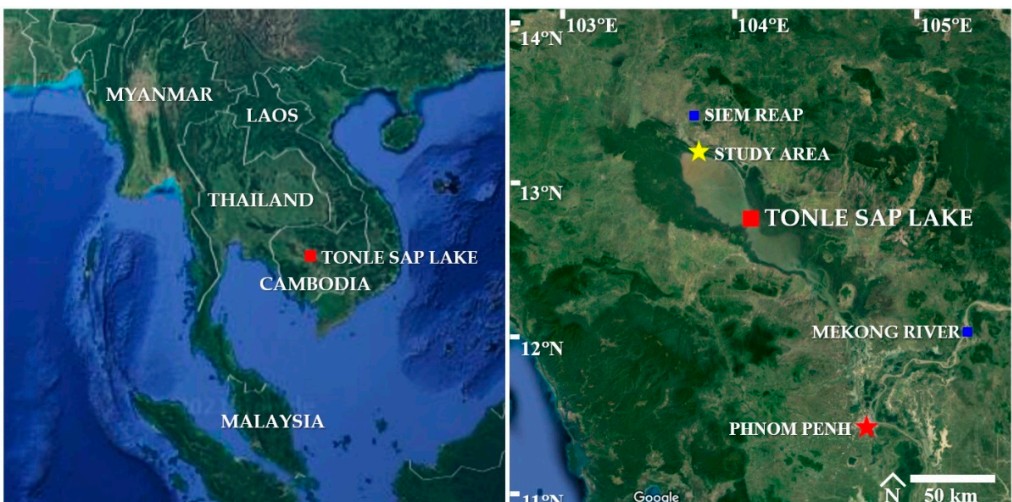

**Figure 1.** Location of the study area situated in the Northwestern part of Tonle Sap Lake, Cambodia (13°23′ N, 103°82′ E) (Source: Google Maps).

### 2.2. Sampling of Sedimentary Water Samples

Sedimentary water samples were collected from the littoral zone of the lake during the late rainy season in November 2018. The definition of sedimentary water samples in this study is the suspended-sediment water samples. The littoral zone is a shallow-water region of the lake that is near the shore area of a terrestrial ecosystem, and it interfaces with the deeper pelagic zone of the aquatic ecosystem [43]. Twenty samples of sedimentary water were taken randomly at depths of 0–10 cm to culture aerobic heterotrophic bacteria. Temperatures of sedimentary water were measured at the site by an Extech 39240 needle probe thermometer (Extech Instruments, Waltham, MA, USA). The pH and salinity values of water samples were measured using ST20M-C Starter Pen Meter (Ohaus, Parsippany, NJ, USA). The collected samples were stored at 4 °C before their analysis, which was performed within 48 h of sampling.

### 2.3. Isolation, Screening, and Purification of Cellulolytic Bacteria from the Sedimentary Water Samples

Sedimentary water samples were serially diluted with sterilized 0.85% (*w/v*) NaCl solution (Sigma-Aldrich, Steinheim, Germany) supplemented with 0.1% (*w/v*) buffered peptone water (Merck, Ponda, India) to obtain 1:10,000 dilutions. One hundred microliters of diluted samples were spread-plated on a growth medium, soyabean casein digest agar (HiMedia, Mumbai, India). The pH value of the growth medium was adjusted to 6.05 (the average pH value of the water samples). The fungicidal nystatin (Alfa Aesar, Lancashire, UK), at a concentration of 50 mg/L, was added to the medium for inhibiting any growths of fungi and yeasts. All culture plates were incubated at 30.5 °C (the average temperature of the sampling site) for 48 h under aerobic conditions in an incubator cabinet (BINDER, Tuttlingen, Germany). The aquatic bacteria were selected based on the investigation of colony morphology and the Gram staining method (Grams Stain Kit, HiMedia, Mumbai, India). They were subsequently screened and purified by streaking on plates of a selective medium for the screening of cellulolytic bacteria, CMC agar. The CMC agar, previously described by Chantarasiri [20], contained 0.2 g of $NaNO_3$ (Alfa Aesar, Lancashire, UK), 0.1 g of $K_2HPO_4$ (Ajax Finechem, Riverwood, New South Wales, Australia), 0.05 g of $MgSO_4$ (Ajax Finechem, New South Wales, Australia), 0.05 g of KCl (Ajax Finechem, New South Wales, Australia), 0.02 g of peptone powder (HiMedia, Mumbai, India), 0.2 g of carboxymethyl cellulose (CMC) (Alfa Aesar, Lancashire, UK) as a sole carbon source, 1.7 g of agar powder (HiMedia, Mumbai, India), and 0.05 mg of nystatin in 100 mL of distilled water. All purified cellulolytic isolates were genetically identified by molecular methods and 16S rRNA gene sequence analysis.

### 2.4. Genetic Identification and Phylogenetic Analysis of Isolated Bacteria

The genomic DNA of each isolated strain was extracted by a genomic DNA isolation kit (Bio-Helix, Keelung City, Taiwan), according to the standard protocol described by Bio-Helix. Polymerase chain reaction (PCR) of the 16S rRNA genes was carried out using the OnePCR reaction mixture (Bio-Helix, Keelung City, Taiwan) with a 27F primer of 5′-AGAGTTTGATCMTGGCTCAG-3′ and a 1492R primer of 5′-TACGGYTACCTTGTTACGA CTT-3′ (Macrogen Inc., Seoul, Korea) [44]. The amplification parameters were adapted from Boontanom and Chantarasiri [45] for 35 amplification cycles in a Mastercycler Nexus Gradient thermal cycler (Eppendorf, Hamburg, Germany). In brief, these amplifications were performed with a preheating step at 94 °C for 4 min, a denaturation step at 94 °C for 40 s, an annealing step at 55 °C for 1 min, an extension step at 72 °C for 1 min and 10 s, followed by a final extension step at 72 °C for 10 min. Approximately 1500-bp PCR products from the amplifications were electrophoresed on a 1.5% (*w/v*) OmniPur agarose gel (Calbiochem, Darmstadt, Germany) and subsequently visualized by Novel Juice (Bio-Helix, Keelung City, Taiwan). The PCR products were cleaned up and nucleotide sequenced using the services of Macrogen Inc. (Seoul, Korea). Finally, they were aligned for identification by the BLASTn program based on the nucleotide collection (nr/nt) database and a megablast algorithm of the National Center for Biotechnology Information (NCBI) [46,47]. The phylogenetic tree of all isolated bacteria was generated by the BIONJ algorithm [48] with 100,000 bootstrap replications using SeaView software version 5.0.2 [49], and visualized by FigTree software version 1.4.4 (Institute of Evolutionary Biology, University of Edinburgh, Edinburgh, UK).

All nucleotide sequences of 16S rRNA gene from this study were deposited in the GenBank database of NCBI [50] under the accession numbers MW344078, MW344081, MW344082, MW344088, MW344090, MW344117, MW344118, MW344120, MW344123, MW344263, MW344267, MW344273, MW344274, and MW344292.

### 2.5. Determination of Hydrolysis Capacity of Isolated Bacteria

The determination of hydrolysis capacity (HC) procedures for the isolated bacteria was previously described by Chantarasiri [20,27]. The isolated bacteria were cultured in soyabean casein digest broth (HiMedia, Mumbai, India) at 30.5 °C for 16 h. One drop (5 μL) of each bacterial growth culture in the soyabean casein digest broth was spot-plated on CMC agar. All inoculated plates were incubated at 30.5 °C for 48 h and then flood-plated with an iodine solution for 15 min. The iodine solution contained 0.33% (*w/v*) $I_2$ (Ajax Finechem, New South Wales, Australia) and 0.67% (*w/v*) KI (Ajax Finechem, New South Wales, Australia). The cellulolytic bacteria were able to produce the hydrolysis zone around their colonies on CMC agar through their extracellular cellulases. The hydrolysis zone was visualized after the iodine solution staining that determined the cellulolytic performance of bacteria. The HC value was calculated by the ratio between the diameter of the hydrolysis zone around the bacterial colony and the diameter of the bacterial colony. All experiments were performed in triplicate.

### 2.6. Preparation of the Crude Cellulases

The cellulolytic bacteria were inoculated in CMC broth contained in baffled flasks (Schott-Duran, Mainz, Germany) for the production of cellulases. All bacteria were cultured under aerobic conditions at 30.5 °C for 48 h in an orbital shaking incubator at 150 rpm (LSI-3016R, Daihan Labtech, Gagokri, Korea). The cultured broths were centrifuged by Digicen 21 R centrifugation (Ortoalresa, Madrid, Spain) at $4500 \times g$ at 4 °C for 30 min to obtain cell-free supernatants. The supernatants were concentrated by 10 kDa Amicon ultra centrifugal filter units (Millipore, Cork, Ireland), and the concentrated supernatants served as crude cellulases extracts, and were kept at 4 °C.

### 2.7. Determination of Cellulolytic Activity of the Crude Cellulases

The cellulolytic activity of the crude cellulases was determined as conducted from the methods described by Chantarasiri [20,27]. All crude cellulases were assayed for

four cellulolytic activities, including endoglucanases, exoglucanases, β-glucosidases, and total cellulases.

The activity of endoglucanases (CMCase) was measured by incubating 0.5 mL of crude cellulases with 2% (*w/v*) CMC (Alfa Aesar, Lancashire, UK) as substrate in 0.5 mL of an assay buffer at 50 °C for 30 min. The activity of exoglucanases (Avicelase) was measured by incubating 0.5 mL of crude cellulases with 2% (*w/v*) Avicel PH-101 (Sigma-Aldrich, Steinheim, Germany) as substrate in 0.5 mL of an assay buffer at 50 °C for one hour. The activity of total cellulases (FPase) was measured by incubating 0.5 mL of crude cellulases with 0.05 g of grade 1 qualitative filter paper (Whatman, Darmstadt, Germany) as substrate in 1 mL of an assay buffer at 50 °C for one hour. The reducing sugars released from the cellulolytic reactions of the substrates were spectrophotometrically determined by a 3,5-dinitrosalicylic acid (DNS) method at 540 nm [51]. The cellulolytic activity values were calculated by a glucose standard curve. One unit (U) of CMCase, Avicelase, and FPase was defined as the amount of enzyme required to release 1 μmol of the reducing sugars as glucose equivalent per minute under the assay conditions.

The activity of β-glucosidases was measured by incubating 0.5 mL of crude cellulases with 1 mL of 0.1% *p*-nitrophenyl-β-D-glucopyranoside (Sigma-Aldrich, Steinheim, Germany) as substrate in an assay buffer at 50 °C for one hour. The reaction of β-glucosidases was terminated by adding 2 mL of 1 M $Na_2CO_3$ solution (Ajax Finechem, New South Wales, Australia). The *p*-nitrophenol molecules released from the cellulolytic reactions of the substrate were spectrophotometrically determined at 405 nm. The activity values of β-glucosidases were calculated by a *p*-nitrophenol standard curve. One unit (U) of β-glucosidase was defined as the amount of enzyme required to release 1 μmol of *p*-nitrophenol per minute under the assay conditions.

The assay buffer used in this experiment was 50 mM sodium phosphate buffer at a pH of 7.0 (Ajax Finechem, New South Wales, Australia), according to the previously described studies [27,52,53]. The spectrophotometric analyses were performed using an AccuReader microplate reader (Metertech, Taipei, Taiwan). All experiments were performed in triplicate.

### 2.8. Bioinformatic Analysis of the Endoglucanase from the Most Active Cellulolytic Bacterium

The bioinformatic analysis of endoglucanase was examined on the most active cellulolytic bacterium, *B. mojavensis* strain REP303. The bacterium was genomic DNA extracted by a genomic DNA isolation kit, PCR amplified by the OnePCR reaction mixture, and PCR products visualized by the Novel Juice, all of which were based on the previously described experiment. The *endoglucanase* gene was amplified using a forward primer of 5′-CATGCCATGGGCATGAAACGGTCAATTTCTATTTTT-3′ and a reverse primer of 5′-CCGCTCGAGATTGGGTTCTGTTCCCCAAA-3′ (Macrogen Inc., Seoul, Korea) [17]. The PCR amplifications were performed for 35 amplification cycles, with a preheating step at 95 °C for 5 min, a denaturation step at 95 °C for 1 min, an annealing step at 65 °C for 1 min (−0.5 °C per cycle), an extension step at 72 °C for 3 min, and a final extension step at 72 °C for 10 min. Approximately 1400-bp PCR products from the amplifications were electrophoresed on an agarose gel. They were subsequently cleaned up and sequenced by the nucleotide sequencing service of Macrogen Inc. (Seoul, Korea).

The resulting nucleotide sequence of the *endoglucanase* gene was bioinformatically analyzed according to the studies of Ma et al. [17] and Wu et al. [54]. The open reading frame (ORF) of the nucleotide sequence was analyzed by the ORF finder program of NCBI and aligned for homology analysis by the BLASTx program, based on the non-redundant protein sequence (nr) database of NCBI [46,47]. The phylogenetic tree of the putative amino acid sequence was generated by the BIONJ algorithm [48] with 100,000 bootstrap replications using SeaView software version 5.0.2 [49], and visualized by FigTree software version 1.4.4 (Institute of Evolutionary Biology, University of Edinburgh, Edinburgh, UK).

The physiochemical characteristics of the putative endoglucanase, including theoretical isoelectric point (pI) and molecular weight ($M_w$), were predicted using the Compute

pI/M$_w$ tool of the ExPASy bioinformatics resource portal [55]. The modular structure of putative endoglucanase was predicted using the Simple Modular Architecture Research Tool [56,57]. Finally, the simulated structure of putative endoglucanase was analyzed by the SWISS-MODEL program [58].

A nucleotide sequence of gene encoding endoglucanase from this study was deposited in the GenBank database of NCBI [50], under accession number MZ133283.

### 2.9. Enzymatic Characterization of the Cellulases from the Most Active Cellulolytic Bacterium

Enzymatic characterization was determined for the CMCase activity of the crude cellulases produced from the most active endoglucanasic *B. mojavensis* strain, REP303. The study focused on three parameters that affected CMCase activity: temperature, pH, and certain chemical additives. CMCase activity was measured accordingly, as mentioned in the previous experiments. All experiments were performed in triplicate.

The effect of temperature on cellulolytic activity and thermal stability was evaluated accordingly, as mentioned in Chantarasiri [27]. The optimum temperature of CMCase activity was determined at temperatures ranging from 25 °C to 80 °C, at a pH of 7.0 for an assay buffer. Thermal stability was measured by pre-incubating the crude cellulases at temperatures ranging from 25 °C to 80 °C for 24 h, at a pH of 7.0 for an assay buffer, and the relative activity of CMCase was monitored afterwards. The assay buffer was a 50 mM sodium phosphate buffer at a pH of 7.0 (Ajax Finechem, New South Wales, Australia).

The effect of pH on the cellulolytic activity and pH stability was evaluated accordingly, as mentioned in Chantarasiri [27], in the pH-varied buffers at 50 °C. The optimum pH of the CMCase activity was determined in assay buffers comprising 50 mM citrate buffer (pH 4.0–6.0), 50 mM sodium phosphate buffer (pH 6.0–8.0), 50 mM Tris-HCl buffer (pH 8.0–9.0), and 50 mM glycine-NaOH buffer (pH 9.0–10.0) (Ajax Finechem, New South Wales, Australia). pH stability was measured by pre-incubating the crude cellulases in the abovementioned buffer at 50 °C for 24 h, and the relative activity of CMCase was monitored afterwards.

The effect of some chemical additives was measured accordingly, as mentioned by Annamalai et al. [59], Seo et al. [60], and Chantarasiri [27]. Crude cellulase extract was pre-incubated in an assay buffer supplemented with chemical additives at 50 °C for one hour. The thirteen metal ions used as chemical additives in this experiment were Ca$^{2+}$ (as CaCl$_2$), Co$^{2+}$ (as CoCl$_2$), Cu$^{2+}$ (as CuCl$_2$), Fe$^{2+}$ (as FeCl$_2$), Hg$^{2+}$ (as HgCl$_2$), K$^+$ (as KCl), Mg$^{2+}$ (as MgCl$_2$), Mn$^{2+}$ (as MnCl$_2$), Na$^+$ (as NaCl$_2$), Ni$^{2+}$ (as NiCl$_2$), Pb$^{2+}$ (as PbCl$_2$), Sr$^{2+}$ (as SrCl$_2$), and Zn$^{2+}$ (as ZnCl$_2$) (Ajax Finechem, New South Wales, Australia). Ethylene diamine tetraacetic acid (EDTA) disodium salt (Calbiochem, Darmstadt, Germany) was the chelating agent that we used as a chemical additive. All the reagents were supplemented in an assay buffer, with a final concentration of 5 mM. There were six organic solvents used as the chemical additives, comprising acetone, dichloromethane, ethanol, ethyl-acetate, methanol, and *n*-hexane, with a final concentration of 25% (*v/v*) and supplemented in an assay buffer. They were pre-incubated with the crude cellulase extract for four hours under the same conditions as other chemical additives. The relative activity of CMCase was monitored after being incubated with various additives. The assay buffer was a 50 mM sodium phosphate buffer at a pH of 7.0 (Ajax Finechem, New South Wales, Australia).

### 2.10. Statistical Analyses

The statistical analyses of data in this study were performed using R software version 4.0.3 [61]. The normality of data and homogeneity of variances were evaluated by the Anderson–Darling test and Bartlett test, respectively. The multiple comparison analyses were determined by one-way ANOVA followed by Tukey's test with a 95% confidence interval ($p < 0.05$).

## 3. Results

### 3.1. Sampling of Sedimentary Water Samples

Twenty sedimentary water samples were collected from the littoral zone of Tonle Sap Lake in Cambodia during the late rainy season. The environmental characteristics of the water samples were sediment suspended and light-brown colored with a muddy smell. Their chemical characteristics were measured, and they showed an average temperature of $30.5 \pm 0.1$ °C, an average pH value of $6.05 \pm 0.02$, and zero salinity value. All chemical characteristics were used as the growth conditions for bacterial cultivation.

### 3.2. Phenotypic Morphology and Genetic Identification of the Isolated Aquatic Cellulolytic Bacteria

Heterotrophic bacteria ($n = 83$) were grown and isolated from the twenty sedimentary water samples using soyabean casein digest agar. A total of 16 different bacterial isolates were harvested in the CMC agar that harbored CMC as a sole carbon source for selecting the cellulolytic bacteria. However, only 14 bacterial isolates could be cultivated, streak-plated, and purified on the CMC agar. The fourteen bacterial strains were considered as axenic isolates of cellulolytic bacteria. Most purified bacteria had a similar pattern of colony morphology involving white pigmentation and a circular shape. Otherwise, their colonies mainly differed in terms of margin pattern, elevation form, and diameter of the colony. Most aquatic cellulolytic bacteria were rod-shaped bacterial cells with Gram-positive characteristics. Only one isolate was Gram-negative bacteria. The morphology of purified aquatic cellulolytic bacteria, in terms of percentage, is shown in Table 1.

**Table 1.** Percentages of morphology for purified aquatic cellulolytic bacteria ($n = 14$) from the sedimentary water samples.

| Pigmentation (Percentage) | | Shape (Percentage) | | Margin (Percentage) | | Elevation (Percentage) | | Diameter of the Colony (Percentage) | |
|---|---|---|---|---|---|---|---|---|---|
| White | 85.72 | Circular | 85.72 | Entire | 71.43 | Convex | 50.00 | <2.00 mm | 21.43 |
| Pale brown | 7.14 | Irregular | 14.28 | Undulate | 28.57 | Raised | 50.00 | 2.00–4.00 mm | 71.43 |
| Orange | 7.14 | Other | 0.00 | Other | 0.00 | Other | 0.00 | >4.00 mm | 7.14 |
| Total | 100.00 | Total | 100.00 | Total | 100.00 | Total | 100.00 | Total | 100.00 |

The aquatic cellulolytic bacteria were genetically identified according to PCR amplification and analysis of their 16S rRNA gene sequences. Nucleotide alignment of the 16S rRNA genes using the BLASTn program revealed that the aquatic cellulolytic bacteria belonged to three genera, including genus *Aeromonas* of the Phylum Proteobacteria, genus *Bacillus* of the Phylum Firmicutes, and genus *Exiguobacterium* of the Phylum Firmicutes, with 99% query coverage and 98–99% identity. All resulting E values obtained from the BLASTn alignments were zero. Twelve isolates of aquatic cellulolytic bacteria were closely similar to various strains of *Bacillus amyloliquefaciens*, *B. cereus*, *B. flexus*, *B. megaterium*, and *B. mojavensis*. The two others were closely similar to *Aeromonas jandaei* strain 4pM28 and *Exiguobacterium indicum* strain DSAM 62. The genetic identification of aquatic cellulolytic bacteria is shown in Table 2. A circular phylogenetic tree of aquatic cellulolytic bacteria using the BIONJ algorithm with 100,000 bootstrap replications was established, as shown in Figure 2.

The bacterial isolates REP302, REP303, and REP306 fell into the same clade of phylogenetic tree clustered with *B. amyloliquefaciens* and *B. mojavensis* (bootstrap value > 95). The REP401, REP402, and REP406 isolates were phylogenetically clustered with *B. cereus* (bootstrap value > 75). The isolate REP901 was grouped in the same clade of *B. flexus* (bootstrap value = 90). Interestingly, the isolate REP304 and isolate REP902 fell into one branch and were phylogenetically clustered with *B. megaterium* strain F4-2-27 (bootstrap value > 90), according to the results from BLASTn alignment. These isolates were clustered in the same phylogenic group of the other three isolates involving isolates REP301, REP305, and REP403 (bootstrap value > 95). Therefore, the predominant aquatic cellulolytic bacteria in this study were the group of *B. megaterium*. The other clades of this phylogenetic tree were *Exiguobacterium* clade and *Aeromonas* clade. Bacterial isolate REP307

was phylogenetically clustered in the clade of *Exiguobacterium* (bootstrap value = 100) that confidently shared a common ancestor with all clades of the *Bacillus* species. The bacterial isolate REP404 was clustered in the *Aeromonas* clade (bootstrap value = 100), which is considered a unique clade of this circular phylogenetic tree.

**Table 2.** Identity percentages of 16S rRNA gene sequences of the 14 cellulolytic bacteria with closely related bacteria.

| Bacterial Isolate | Closely Related Bacteria | GenBank Accession Number (References) | Query Cover (%) | Identity (%) * | GenBank Accession Number (Deposited) |
|---|---|---|---|---|---|
| REP301 | *Bacillus megaterium* strain WH13 | MF431767.1 | 99 | 99.04 | MW344078 |
| REP302 | *Bacillus amyloliquefaciens* strain SA70 | MK467609.1 | 99 | 98.97 | MW344081 |
| REP303 | *Bacillus mojavensis* strain UMF29 | MG192312.1 | 99 | 98.66 | MW344082 |
| REP304 | *Bacillus megaterium* strain F4-2-27 | MN190174.1 | 99 | 98.50 | MW344088 |
| REP305 | *Bacillus megaterium* strain WBJR0203 | MN993647.1 | 99 | 98.82 | MW344090 |
| REP306 | *Bacillus amyloliquefaciens* strain R2.3 | LC414175.1 | 99 | 98.61 | MW344118 |
| REP307 | *Exiguobacterium indicum* strain DSAM 62 | MH819520.1 | 99 | 98.72 | MW344117 |
| REP401 | *Bacillus cereus* strain BBS15 | MK956956.1 | 99 | 99.09 | MW344120 |
| REP402 | *Bacillus cereus* strain LXJ77 | MN746190.1 | 99 | 98.35 | MW344123 |
| REP403 | *Bacillus megaterium* strain ANA29 | MT122820.1 | 99 | 99.06 | MW344267 |
| REP404 | *Aeromonas jandaei* strain 4pM28 | FJ940804.1 | 99 | 98.78 | MW344263 |
| REP406 | *Bacillus cereus* strain KUBOTAB5 | MK855405.1 | 99 | 98.12 | MW344274 |
| REP901 | *Bacillus flexus* strain Xh8 | MK012676.1 | 99 | 99.10 | MW344273 |
| REP902 | *Bacillus megaterium* strain F4-2-27 | MN190174.1 | 99 | 98.13 | MW344292 |

* Identity results were analyzed on 1 December 2020.

The cellulolytic bacteria were found to be closely related based on the sequence alignment results of the 16S rRNA gene when the identity was more than 98%, such as *B. megaterium* strain REP301. All nucleotide sequences of 16S rRNA genes obtained from this study were deposited in the GenBank database of NCBI under accession numbers MW344078, MW344081, MW344082, MW344088, MW344090, MW344117, MW344118, MW344120, MW344123, MW344263, MW344267, MW344273, MW344274, and MW344292. The aquatic cellulolytic bacteria in this study were stored as frozen stocks in 20% (*v/v*) glycerol and kept at the Faculty of Science, Energy, and Environment, King Mongkut's University of Technology North Bangkok, Thailand.

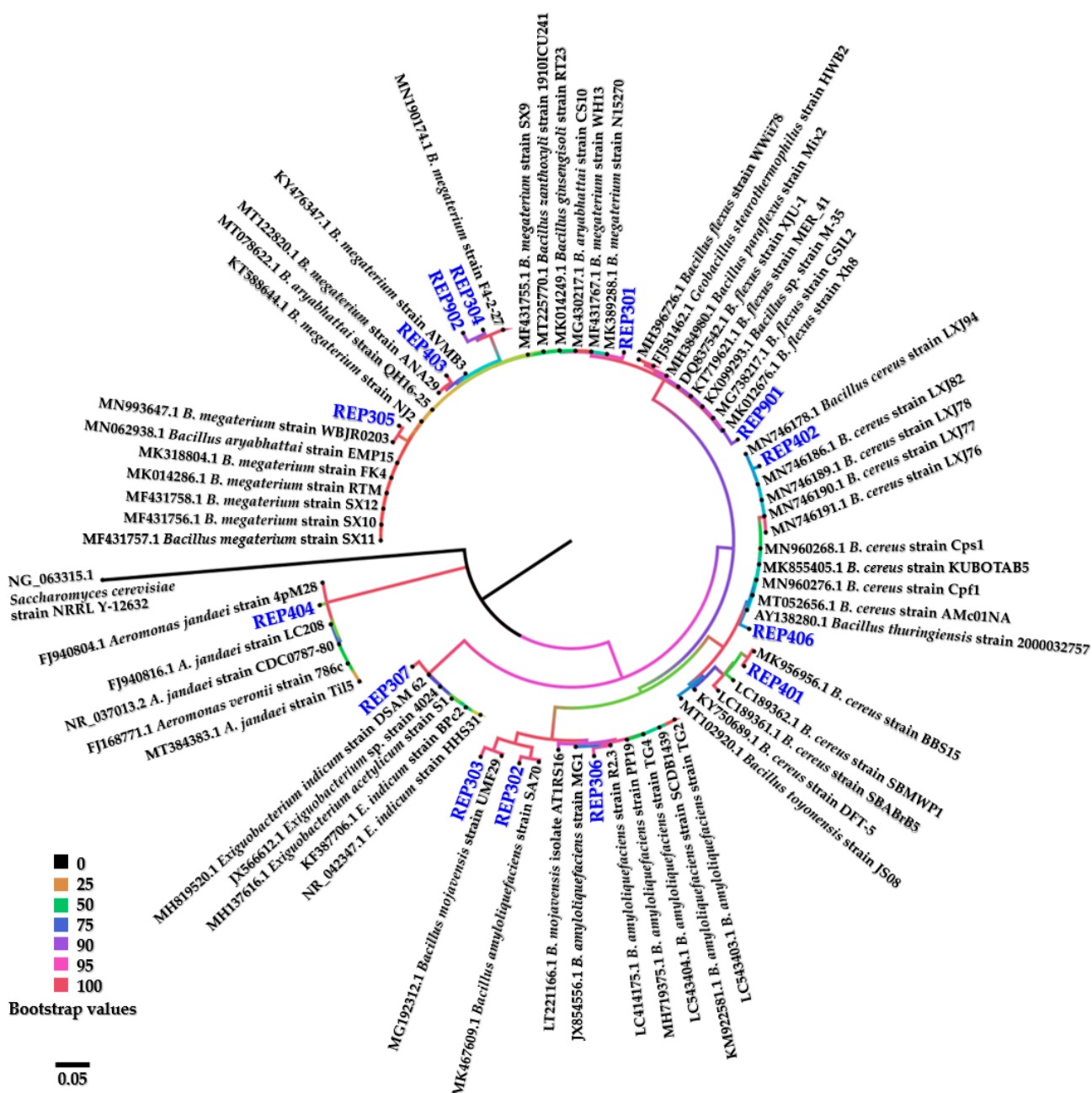

**Figure 2.** Circular phylogenetic tree of aquatic cellulolytic bacteria shown using the BIONJ algorithm with 100,000 bootstrap replications. The phylogenetic tree was generated by SeaView software version 5.0.2 and visualized by FigTree software version 1.4.4.

### 3.3. Hydrolysis Capacity (HC) and Cellulolytic Activity of Isolated Aquatic Cellulolytic Bacteria

The HC values of aquatic cellulolytic bacteria were determined on the CMC agar plates that were flood-plated with iodine solution. This method is used to evaluate the primary enzymatic activity of the active cellulolytic bacteria. The HC values of the fourteen bacterial strains ranged from $1.87 \pm 0.02$ to $5.14 \pm 0.26$. The results showed that *B. cereus* strain REP406 had a significant maximum HC of $5.14 \pm 0.26$ ($p < 0.01$). The next most effective bacteria based on HC values were *A. jandaei* strain REP404 and *E. Indicum* strain REP307. The cellulolytic zone around the bacterial colonies on the CMC agar plates after iodine staining is shown in Figure 3. The HC values of aquatic cellulolytic bacteria are shown in Table 3.

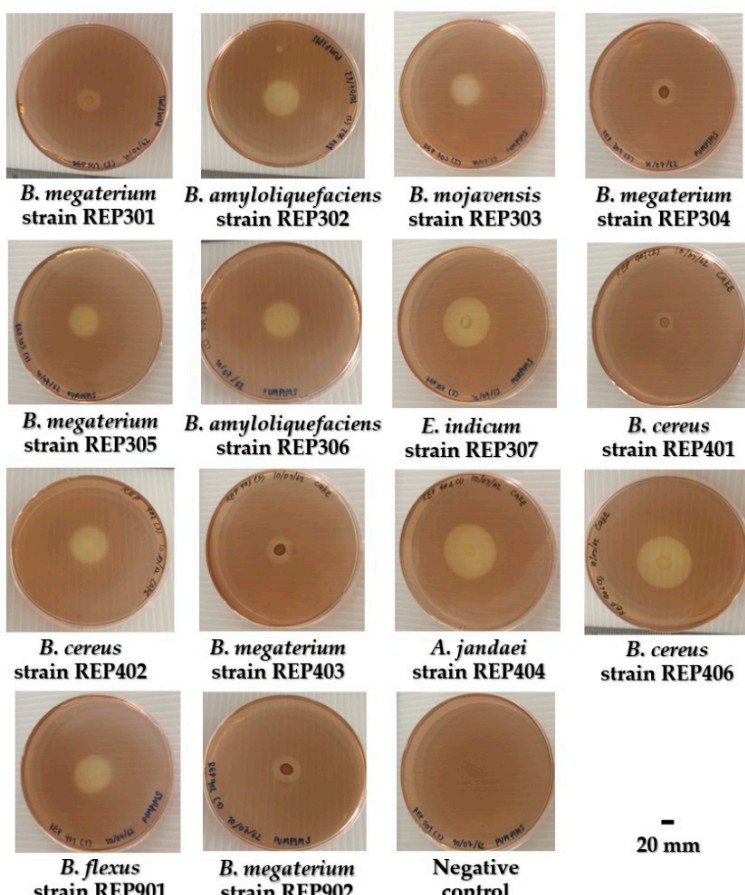

**Figure 3.** The cellulolytic zone around the bacterial colonies of aquatic cellulolytic bacteria on the CMC agar plates after iodine staining. The negative control was non-cellulolytic bacteria inoculated on the CMC agar.

**Table 3.** HC values and enzyme activity of fourteen aquatic cellulolytic bacteria.

| Bacterial Strain | HC Value | Cellulolytic Activity (U/mL) | | | |
|---|---|---|---|---|---|
| | | CMCase | Avicelase | β-Glucosidases | FPase |
| *B. megaterium* strain REP301 | 1.88 ± 0.07 [a] | 2.45 ± 0.05 [bc] | 1.27 ± 0.08 [ab] | 0.13 ± 0.01 [ab] | 1.23 ± 0.01 [bc] |
| *B. amyloliquefaciens* strain REP302 | 3.39 ± 0.18 [d] | 3.26 ± 0.06 [e] | 1.50 ± 0.19 [ac] | 0.13 ± 0.01 [ab] | 1.48 ± 0.04 [f] |
| *B. mojavensis* strain REP303 | 3.19 ± 0.06 [cd] | 3.97 ± 0.05 [g] | 1.94 ± 0.11 [c] | 0.16 ± 0.01 [b] | 1.76 ± 0.05 [h] |
| *B. megaterium* strain REP304 | 2.25 ± 0.18 [ab] | 2.36 ± 0.04 [ab] | 1.28 ± 0.27 [ab] | 0.12 ± 0.00 [ab] | 1.16 ± 0.01 [ab] |
| *B. megaterium* strain REP305 | 2.67 ± 0.28 [bc] | 2.43 ± 0.02 [bc] | 1.66 ± 0.22 [bc] | 0.12 ± 0.01 [ab] | 1.19 ± 0.02 [ac] |
| *B. amyloliquefaciens* strain REP306 | 3.17 ± 0.04 [cd] | 3.72 ± 0.07 [f] | 1.68 ± 0.20 [bc] | 0.13 ± 0.01 [ab] | 1.66 ± 0.05 [g] |
| *E. indicum* strain REP307 | 4.44 ± 0.25 [e] | 2.31 ± 0.06 [ab] | 1.25 ± 0.02 [ab] | 0.11 ± 0.00 [a] | 1.18 ± 0.01 [ac] |
| *B. cereus* strain REP401 | 1.87 ± 0.02 [a] | 2.79 ± 0.15 [d] | 1.30 ± 0.13 [ab] | 0.13 ± 0.00 [ab] | 1.36 ± 0.02 [e] |
| *B. cereus* strain REP402 | 3.55 ± 0.27 [d] | 2.69 ± 0.05 [d] | 1.41 ± 0.13 [ab] | 0.12 ± 0.01 [ab] | 1.33 ± 0.02 [de] |
| *B. megaterium* strain REP403 | 2.30 ± 0.13 [ab] | 2.20 ± 0.06 [a] | 1.02 ± 0.07 [a] | 0.12 ± 0.01 [ab] | 1.12 ± 0.02 [a] |
| *A. jandaei* strain REP404 | 4.72 ± 0.20 [ef] | 2.71 ± 0.03 [d] | 1.20 ± 0.15 [ab] | 0.11 ± 0.00 [a] | 1.12 ± 0.01 [a] |
| *B. cereus* strain REP406 | 5.14 ± 0.26 [f] | 2.61 ± 0.03 [cd] | 1.26 ± 0.12 [ab] | 0.13 ± 0.01 [ab] | 1.26 ± 0.05 [cd] |
| *B. flexus* strain REP901 | 3.53 ± 0.31 [d] | 2.34 ± 0.10 [ab] | 1.31 ± 0.31 [ab] | 0.11 ± 0.00 [a] | 1.15 ± 0.04 [ab] |
| *B. megaterium* strain REP902 | 2.28 ± 0.15 [ab] | 2.36 ± 0.07 [ab] | 1.12 ± 0.18 [a] | 0.11 ± 0.00 [a] | 1.17 ± 0.02 [ab] |

Note: All data showed normality, as evaluated by the Anderson–Darling test. All groups of data had homogeneity of variance, as evaluated by the Bartlett test. The mean values in the same column followed by the same letter were not significantly different among the bacteria, according to Tukey's test. Statistical analyses were determined with a 95% confidence interval. All experiments were performed in triplicate.

The aquatic cellulolytic bacteria were examined for cellulolytic activity assays consisting of endoglucanase (CMCase), exoglucanase (Avicelase), β-glucosidase, and total

cellulase (FPase) activities. The crude cellulases harvested from those bacteria were used for cellulolytic examination. The cellulolytic activity assays showed that they could produce crude cellulases with $2.20 \pm 0.06$ to $3.97 \pm 0.05$ U/mL of the CMCase activity, $1.02 \pm 0.07$ to $1.94 \pm 0.11$ U/mL of the Avicelase activity, $0.11 \pm 0.00$ to $0.16 \pm 0.01$ U/mL of the β-glucosidase activity, and $1.12 \pm 0.01$ to $1.76 \pm 0.05$ of the FPase activity.

*B. mojavensis* strain REP303 was considered the most active cellulolytic bacterium with significant cellulolytic activity among the fourteen bacterial strains ($p < 0.01$). All bacterial strains satisfactorily produced CMCases but produced barely any β-glucosidase. Therefore, the aquatic cellulolytic bacteria in this study were designated as endoglucanasic bacteria. Their cellulolytic activities are shown in Table 3.

### 3.4. Bioinformatic Analysis of the Endoglucanase from B. mojavensis Strain REP303

The bioinformatic analysis of endoglucanase was performed on the most active cellulolytic bacterium, *B. mojavensis* strain REP303. Its *endoglucanase* gene was PCR amplified and analyzed for the nucleotide sequence. The resulting 1400 bp-nucleotide sequences were bioinformatically analyzed for the open reading frame (ORF) of protein-encoding segments. The results found that the amplified *endoglucanase* gene of *B. mojavensis* strain REP303 had a possible ORF of a protein-encoding segment. It could be encoded as a putative protein consisting of 359 amino acid residues. The alignment for homology analysis using the BLASTx program revealed that this *endoglucanase* gene encoded an enzyme with homology of cellulase in the glycosyl hydrolase (GH) family from *Bacillus velezensis* (NCBI reference sequence: WP_095315422.1) with 97% query coverage and 97.71% identity. The E value of the BLASTx alignment result was zero. A cladogram phylogenetic tree of putative endoglucanase from *B. mojavensis* strain REP303 was established using the BIONJ algorithm with 100,000 bootstrap replications, as shown in Figure 4.

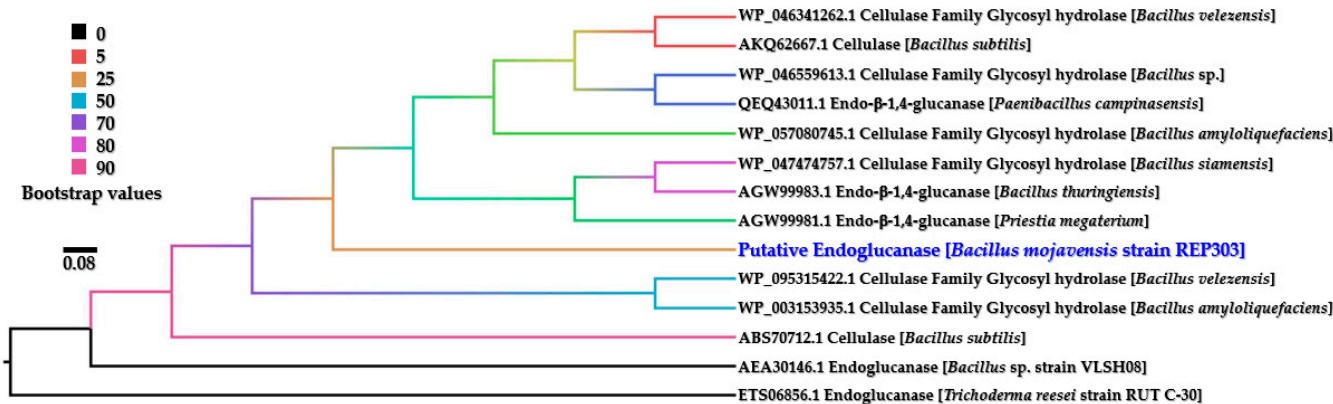

**Figure 4.** Phylogenetic cladogram of putative endoglucanase from *B. mojavensis* strain REP303 using the BIONJ algorithm with 100,000 bootstrap replications. The phylogenetic tree was generated by SeaView software version 5.0.2 and visualized by FigTree software version 1.4.4.

Noticeably, the putative endoglucanase from *B. mojavensis* strain REP303 fell into the same clade of phylogenetic tree that clustered with cellulases in glycosyl hydrolase family and endo-β-1,4-glucanases (endoglucanases) from various *Bacillus* species, such as *B. amyloliquefaciens*, *B. siamensis*, *B. subtilis*, *B. thuringiensis* and *B. velezensis* (bootstrap value = 25). The nucleotide sequence of the *endoglucanase* gene obtained from this study was deposited in the GenBank database of NCBI under accession number MZ133283.

The physiochemical characteristics of the putative endoglucanase from *B. mojavensis* strain REP303 were bioinformatically predicted using the Compute pI/$M_w$ tool. The predicted results showed that its theoretical isoelectric point (pI) and molecular weight ($M_w$) were 6.05 and 39.78 kDa, respectively. The modular structure of this putative endoglucanase was predicted using the Simple Modular Architecture Research Tool. It was found that the catalytic domain consisted of 82 amino acid residues located from Valine 223 to Glycine

304 of the enzyme structure. Its catalytic domain was classified in the carbohydrate-binding module family 3 (CBM3) of glycosyl hydrolase family 5 (GH5). This catalytic domain was similar to the CBM3 of cellulolytic *B. subtilis* (EMBL-EBI reference number SP: P07983), with 96.3% identity. The three-dimensional structure simulation, using the SWISS-MODEL program, is shown in Figure 5. The template for the three-dimensional structure simulation was the endo-1,4-β-glucanase from *B. subtilis* (PDB: 3pzt) [62], which was selected automatically by the SWISS-MODEL program. The scores of a simulated three-dimensional structure were 1.16 for MolProbity Score, 0.34 for Clash Score, 96.35% for Ramachandran Favored, 0.00% for Ramachandran Outliers, and 2.45% for Rotamer Outliers. Unfortunately, the binding domain and several three-dimensional structures in the N-terminal domain of the putative endoglucanase could not be bioinformatically simulated for *B. mojavensis* strain REP303.

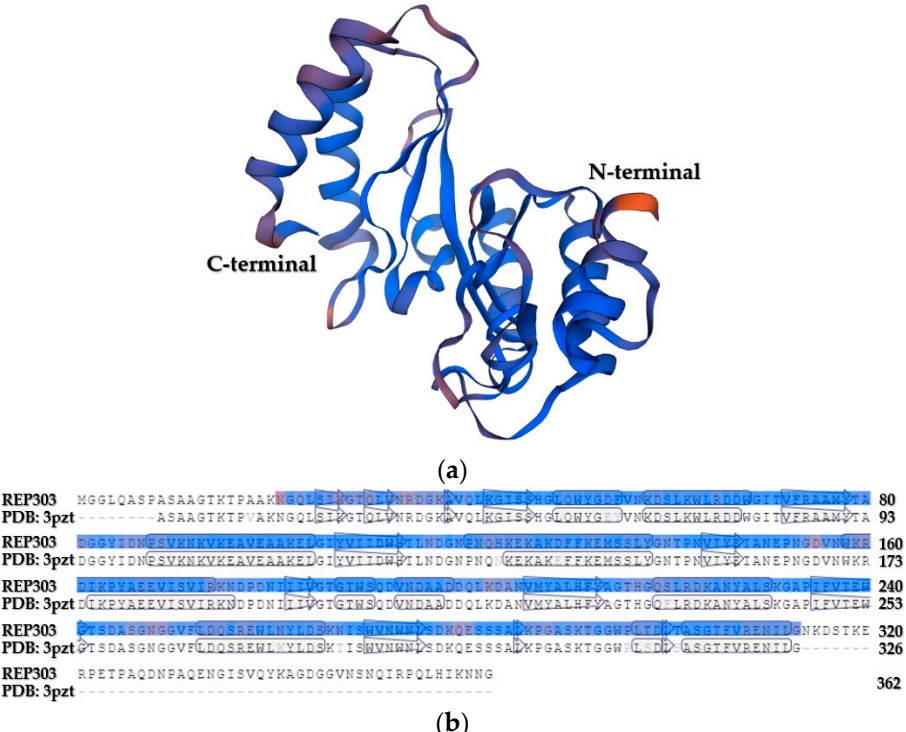

(**a**)

(**b**)

**Figure 5.** (**a**) Simulated three-dimensional structure of putative endoglucanase from *B. mojavensis* strain REP303. The enzyme structure was generated by the SWISS-MODEL program based on the structure of the endo-1,4-β-glucanase from *B. subtilis* (PDB: 3pzt). (**b**) The SWISS-MODEL alignment among a model (REP303 as the putative endoglucanase from *B. mojavensis* strain REP303) and a template (PDB: 3pzt as the endo-1,4-β-glucanase from *B. subtilis* [62]).

### 3.5. Enzymatic Characterization of Cellulases from B. mojavensis Strain REP303

Crude cellulases from *B. mojavensis* strain REP303 were characterized for CMCase activity at different temperatures, pH values, and chemical additives. The optimum temperature and pH for the enzyme are shown in Figures 6a and 7a, respectively. Interestingly, the buffer types significantly affected CMCase activity at the same pH value after one hour of experimental incubation, as shown in Figure 7a. The optimum temperature and pH for the CMCase activity were a moderate temperature of 50 °C ($p < 0.01$) with a neutral pH ranging from 7.0 to 8.0 of sodium phosphate buffer ($p < 0.01$).

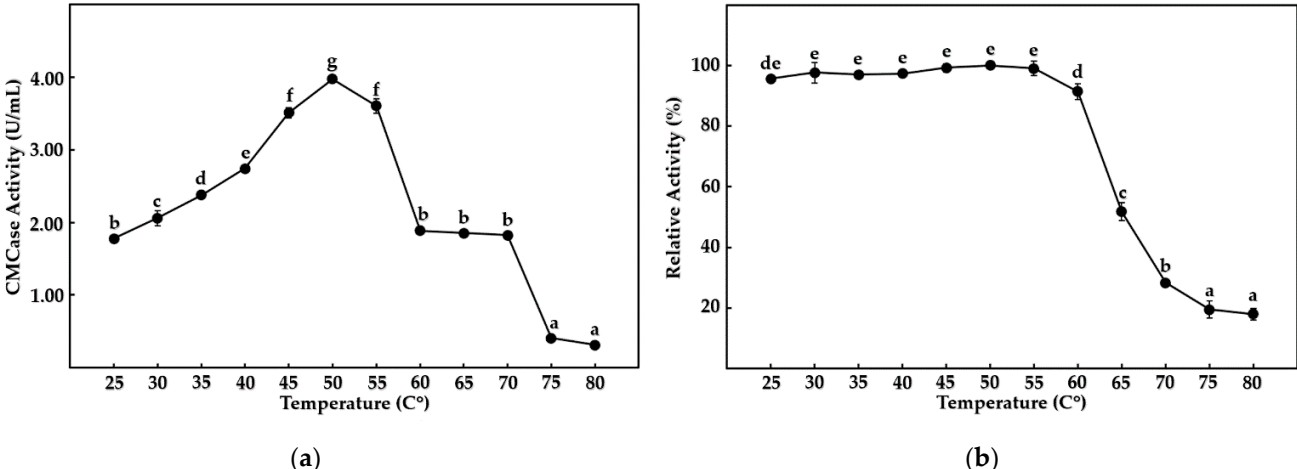

**Figure 6.** Effect of temperature on CMCase activity from *B. mojavensis* strain REP303, with (**a**) showing optimum temperature and (**b**) showing thermal stability. Error bars represent the standard deviation of the three replicates. The mean values followed by the same letter were not significantly different in terms of CMCase activity, according to Tukey's test ($p < 0.05$).

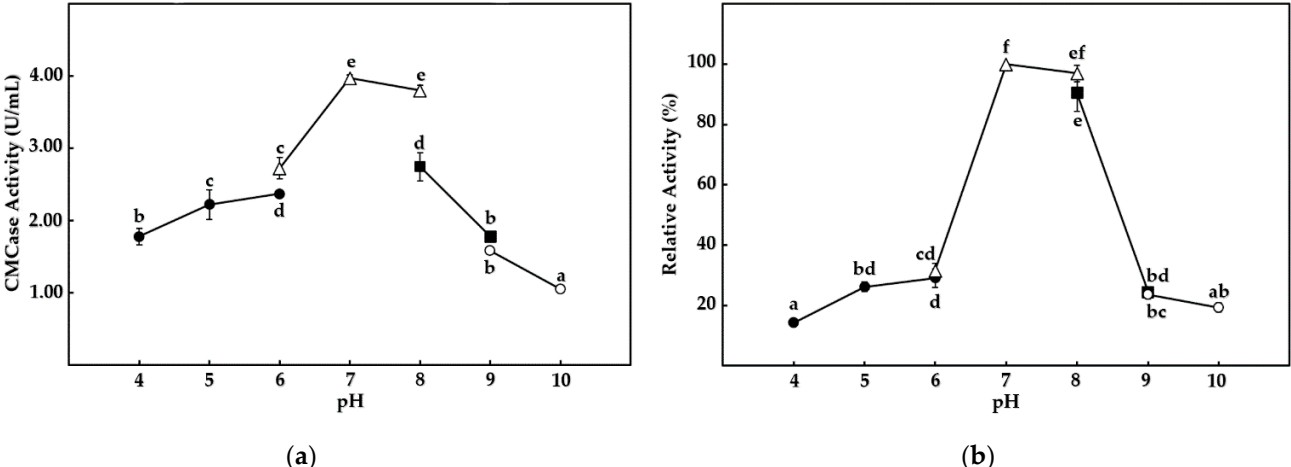

**Figure 7.** Effect of pH values on CMCase activity from *B. mojavensis* strain REP303, with (**a**) showing optimum pH and (**b**) showing pH stability. CMCase activity was measured in citrate buffer (●), sodium phosphate buffer (△), Tris-HCl buffer (■), and glycine-NaOH buffer (○). Error bars represent the standard deviation of the three replicates. The mean values followed by the same letter were not significantly different in terms of CMCase activity, according to Tukey's test ($p < 0.05$).

The enzyme activity remained stable up to 55 °C ($p < 0.01$), while the activity was reduced to below 50% of relative CMCase activity after incubation at a temperature above 65 °C for 24 h. Thermal and pH stability values for the enzyme are shown in Figures 6b and 7b, respectively. Different buffers with the same pH were not significantly affected by the stability of CMCase with prolonged incubation, as shown in Figure 7b. The enzyme was stable by above 90% of relative activity ($p < 0.01$) at a neutral pH, ranging from 7.0 to 8.0, after incubation for 24 h. The enzyme was nearly inactivated at high temperature, or a high acidic content, and had strong basic conditions, with less than 20% relative activity remaining.

The effects of various chemical additives on the CMCase activity of crude cellulases from *B. mojavensis* strain REP303 are shown in Table 4. The results revealed that most metal ions did not both inhibit and enhance the CMCase activity of the enzyme under optimal enzymatic conditions. The enzyme was significantly enhanced by $Co^{2+}$, $Pb^{2+}$, and $Mn^{2+}$ at a final concentration of 5 mM with 133%, 137%, and 162% of relative activity, respectively ($p < 0.01$). A metal ion chelating agent, EDTA, could significantly inhibit

CMCase activity, with 45% of relative activity remaining ($p < 0.01$). Five organic solvents were considered weak inhibitors, with a final concentration of 25% ($v/v$). They could barely inhibit CMCase activity after four hours of incubation. However, ethyl-acetate significantly inhibited enzyme activity at the same quantity of EDTA. It could inhibit CMCase activity, with 33% of relative activity remaining ($p < 0.01$).

**Table 4.** Effect of various chemical additives on CMCase activity from *B. mojavensis* strain REP303.

| Chemical Additives | Relative Activity (%) |
|:---:|:---:|
| $Ca^{2+}$ | 102.58 ± 2.44 [d] |
| $Co^{2+}$ | 133.48 ± 5.36 [e] |
| $Cu^{2+}$ | 102.73 ± 3.32 [d] |
| $Fe^{2+}$ | 104.75 ± 6.56 [d] |
| $Hg^{2+}$ | 105.88 ± 6.14 [d] |
| $K^+$ | 98.98 ± 2.18 [cd] |
| $Mg^{2+}$ | 95.54 ± 1.80 [bd] |
| $Mn^{2+}$ | 161.62 ± 2.40 [f] |
| $Na^+$ | 98.83 ± 3.28 [cd] |
| $Ni^{2+}$ | 95.89 ± 2.43 [bd] |
| $Pb^{2+}$ | 136.80 ± 4.14 [e] |
| $Sr^{2+}$ | 98.72 ± 1.96 [cd] |
| $Zn^{2+}$ | 100.95 ± 3.51 [d] |
| EDTA | 44.51 ± 4.90 [a] |
| Acetone | 87.97 ± 5.06 [bc] |
| Dichloromethane | 99.35 ± 0.51 [cd] |
| Ethanol | 88.06 ± 1.46 [bc] |
| Ethyl-acetate | 33.23 ± 3.95 [a] |
| Methanol | 85.89 ± 3.32 [b] |
| *n*-Hexane | 98.90 ± 3.55 [cd] |

Note: All data showed normality, as evaluated by the Anderson–Darling test. All groups of data had homogeneity of variances, as evaluated by the Bartlett test. The mean values followed by the same letter were not significantly different among the CMCase activities, according to Tukey's test. Statistical analyses were determined with a 95% confidence interval. All experiments were performed in triplicate.

## 4. Discussion

Water samples were collected from the littoral zone of Tonle Sap Lake in Cambodia to study the diversity and cellulolytic activity of culturable cellulolytic bacteria. The lake water is a deep-brown color all year round due to the tremendous amount of muddy suspended sediments [63]. Sediments in the lake are driven by an annual flood pulse phenomenon between the lake and the Mekong River [64]. Therefore, the morphology of all water samples in this study was suspended and muddy-smelling sediment. The average temperature and pH of collected water samples were 30.5 °C and 6.05, respectively; these values are similar to previous studies concerning temperature and pH of samples in Tonle Sap Lake, for which the temperature was 26–35 °C and the pH was 6.5–8.9 [41,63].

The littoral zone of the lake is composed of various particles, with sizes ranging from fine sediments to boulders. Lake sediments are an important storage site for organic matter and microorganisms [29]. Such organic matter generally originates from algae, aquatic plants, terrestrial plant litters, and human waste, which are naturally decomposed by various phyla of aquatic bacteria and archaea through hydrolytic enzymes [6,65]. Accordingly, most bacteria are responsible for contributing to several biogeochemical and energy cycles in lake environments [6].

In this study, there were 83 aquatic bacterial isolates harvested from the sedimentary water samples of Tonle Sap Lake. The screening of cellulolytic bacteria using the CMC agar method showed 14 bacterial isolates (17% of the isolated bacteria) that were designated as cellulolytic bacteria sustainably cultivated in a laboratory. A recent study reported that there were 45 aquatic cellulolytic bacteria (32% of the isolated bacteria) isolated from the water samples of the Dal Lake in India; only 3.6% of the isolated bacteria were considered

highly efficient cellulolytic bacteria [31]. Moreover, a previous study found that only five aquatic cellulolytic bacteria could be screened from a freshwater lake in Tamil Nadu, India [30]. It was believed that the active cellulolytic bacteria were valuable and rarely found in freshwater lake ecosystems.

Molecular identification of the 16S rRNA gene revealed that the fourteen isolated cellulolytic bacteria belonged to genus *Aeromonas* of the Phylum Proteobacteria, genus *Bacillus* of the Phylum Firmicutes, and genus *Exiguobacterium* of the Phylum Firmicutes. Genetic identification was confidently supported by the resulting phylogenetic tree and its bootstrap values. The results suggested that the *Bacillus* species were the predominant culturable cellulolytic bacteria in the lake water samples. This might be because the *Bacillus* species can utilize a variety of substances and survive even in an extreme environment by forming endospores [66]. From December 2016 to August 2017, Tonle Sap Lake was studied in relation to its bacterial community in water and sediment samples collected from the Northwestern and Southeastern parts of the lake [41]. It was shown that the most dominant bacteria phyla were Actinobacteria, Chloroflexi, Cyanobacteria, Firmicutes, and Proteobacteria. Proteobacteria were abundant in the surface water and sub-layer water, while Firmicutes were abundant in the sediment layer of the lake's bottom. Little is known about the common dataset of aquatic cellulolytic bacteria in the littoral zone of Tonle Sap Lake. For freshwater lakes elsewhere, several aquatic cellulolytic bacteria have been isolated and identified. A *Bacillus* sp. strain BF-16 was isolated from lake water samples in India and preliminarily designated as a cellulolytic bacterium [30]. *Bacillus* sp. strain WDHS-01 and *Bacillus* sp. strain WDHS-03 were isolated from the sediment samples of Lake Donghu in China and found to be cellulolytic bacteria with satisfactory endoglucanase activity [29].

The aquatic cellulolytic *Bacillus* species in this study were *B. amyloliquefaciens*, *B. cereus*, *B. flexus*, *B. megaterium*, and *B. mojavensis*. The results revealed that *B. megaterium* was the most abundant cellulolytic species, accounting for 42% isolated *Bacillus* in this study. This might be because *B. megaterium* can cultivate on more than 62 carbon sources in simple media [67]. Several *B. megaterium* strains were isolated from freshwater environments and designated as cellulolytic bacteria [27,34]. The four other *Bacillus* species have been reported as cellulase-producing bacteria in many previous studies. However, the data from these *Bacillus* species isolated from freshwater lakes and related freshwater environments have not been well characterized at this time. There are few aquatic cellulolytic strains that have been reported from marine and mangrove swamp samples. *B. amyloliquefaciens* is a plant-root-associated bacterium that has been found to secrete cellulases [68–70]. This bacterium was also isolated and identified in mangrove sediments in Indonesia [71]. *B. cereus* is also distributed and found in aquatic environments [72]. A previous study reported that aquatic *B. cereus* strain JD0404 from a mangrove swamp in Thailand was an active cellulolytic bacterium, and biodegraded cellulose with its endoglucanase activity [20]. An isolate of *B. flexus* was isolated from degraded seaweed in India, which revealed its alkali-halotolerant cellulase [73]. Lastly, *B. mojavensis* has been isolated from desert soil, medical plants, and rhizosphere samples [74–76]. Only a few strains from soil samples have been confirmed to be cellulolytic bacteria [77,78]. Importantly, this study showed that *B. mojavensis* strain REP303 could be isolated from the sedimentary water samples of Tonle Sap Lake and certainly designated as an active cellulolytic bacterium.

Another aquatic cellulolytic bacterium of the Phylum Firmicutes found in this study was *Exiguobacterium indicum*. Many previous reports showed that the *E. indicum* were isolated from various aquatic environments, such as melted water from glaciers in India [79], a hot spring in Vietnam [80], and a freshwater lake in China [81]. There were some strains of *Exiguobacterium* species that might produce cellulases [82–84]. However, reports on cellulolytic *E. indicum* and their cellulases have been rare. It is surprising that the cellulolytic bacterium isolate REP307 found in the study was closely related to *E. indicum* according to the orange color of a bacterial colony, phenotypic characteristics, genetic alignment of 16S rRNA gene, and a bootstrap value of the phylogenetic tree. Ung et al. [41] suggested that

the Firmicutes bacteria found in Tonle Sap Lake reflected the contamination of lake water from anthropogenic wastes, as well as human feces.

The cellulolytic *Aeromonas jandaei* strain REP404 was the only Gram-negative bacterium of the phylum Proteobacteria that was isolated from the sedimentary water samples in this study. The *A. jandaei* has been isolated from diverse freshwater habitats and has been associated with a wide range of bacterial diseases in humans [85,86] and freshwater fish [87]. Ung et al. [41] expressed concern that the Proteobacteria found in Tonle Sap Lake could pose a potential risk to human health. As reported, three strains of *A. jandaei* were isolated from the guts of herbivorous freshwater fish in China, and showed CMCase, microcrystalline cellulase, and β-glucosidase activities [88].

The most active cellulolytic bacteria based on HC value was *B. cereus* strain REP406. It showed a significant HC value of 5.14, which is higher than that of previously reported *Bacillus* species from other aquatic environments. The aquatic *Bacillus* sp. strain W0105 from a freshwater wetland in Thailand showed a maximum HC value of 3.54 [27]. Aquatic *B. cereus* strain JD0404, isolated from a mangrove swamp in Thailand, exhibited a maximum HC of 4.47 [20]. The most active form of cellulolytic bacteria based on enzymatic activity was *B. mojavensis* strain REP303. It showed the most significant CMCase, Avicelase, β-glucosidase, and FPase activities, as mentioned above in the results. *B. mojavensis* strain REP303 had more CMCase activity than some of the aquatic *Bacillus* isolated from other lakes, such as *Bacillus* sp. strain WDHS-01 and *Bacillus* sp. strain WDHS-03 from Lake Donghu in China [29]. All isolated aquatic bacteria in this study showed satisfactory CMCase activity and little β-glucosidase activity. Therefore, endoglucanases were primarily responsible for the cellulolytic activity that was produced in their cellulolytic mechanisms.

Interestingly, *B. mojavensis* strain REP303 was not the most active cellulolytic bacteria based on HC value. A big concern about the agar plate-screening method, such as the standard CMC agar method, was the conflicting correlation between resulting HC value and enzyme activity [89]. This conflicting result may be due to the fluctuations in some experimental parameters, which affected the cellulase-producing processes described in several previous studies [26,27,34,90]. To solve this problem, the development of novel screening methods should be advanced further.

The most active endoglucanasic *B. mojavensis* strain REP303 was considered a potential candidate for bioinformatic studies. Bioinformatic analyses revealed that the *endoglucanase* gene of *B. mojavensis* strain REP303 could be encoded as a putative endoglucanase with 359 amino acid residues. It was related to a GH family cellulase from *B. velezensis* by the BLASTx program. This putative endoglucanase was classified as an endoglucanase belonging to the GH5 family, which is the largest and most functionally diverse family of endoglucanases detected in numerous species [91–93]. The phylogenetic tree showed that this putative endoglucanase was clustered with the endoglucanases from *B. amyloliquefaciens*, *B. siamensis*, *B. subtilis*, *B. thuringiensis*, and *B. velezensis*. The relationship was not well supported by the bootstrap value. It was suggested that this putative endoglucanase was only related to the *Bacillus* family, but was not encoded from the five mentioned *Bacillus* species. Many studies have been carried out to clone and express the genes encoding for these GH5 cellulases in heterologous hosts for industrial applications [94]. The three-dimensional structure of the putative endoglucanase has been simulated successfully. However, this simulated structure was not considered the optimal structure based on the resulting scores from the SWISS-MODEL program. It should be improved by advanced bioinformatic analyses and X-ray crystallography.

Crude cellulase extracts from *B. mojavensis* strain REP303 were enzymatically characterized and evaluated for biotechnological applications. The CMCase activities of crude cellulase from *B. mojavensis* strain were active and stable under meso-temperature and neutral pH conditions. This was similar to the CMCase activity of other aquatic *Bacillus* species, which were active at a temperature range of 50–55 °C and a pH range of 7.0–8.0 [20,27,95]. The aquatic *Bacillus* cellulases had been reported for stability at a temperature range of 20–60 °C and a pH range of 4.0–8.0 [20,27,95]. However, crude cellulases from *B. mojavensis*

strain REP303 were slightly different from other aquatic *Bacillus* cellulases. This strain was inactivated and unstable at a weakly acidic pH of 4.0–6.0, which correlated with its theoretical pI of 6.05. Enzyme solubility reaches a minimum when the pH of the environment is equal to the pI, resulting in low enzymatic activity [96]. This *B. mojavensis* strain REP303 was preferable for various biotechnological and industrial applications because its cellulases could hydrolyze the cellulose-based materials under mild conditions. The advantages of CMCase activating at a neutral pH and meso-temperature include a lower energy requirement and better environmental friendliness. The suggested biotechnological and industrial applications, based on the active endoglucanase (CMCase) under mild conditions, were the food and feed industry, as well as the pulp and paper industry [97].

The effects of various chemical additives on enzymatic activity from *B. mojavensis* strain REP303 were determined. CMCase activity was found to be significantly enhanced by $Mn^{2+}$. Similarly, many previous reports showed that $Mn^{2+}$ could activate the CMCase activity of aquatic *Bacillus* cellulases [20,27,53,95]. Therefore, it could be promising as an effective activator of CMCase in further biotechnological applications due to it being less toxic and cheaper than other metal ions [27]. The influence of metal ions on cellulase activity suggests that they may interact with basic or acidic amino acid residues of the enzymes [98]. CMCase activity was significantly inhibited by EDTA and ethyl-acetate, which is similar to those sources of inhibition of CMCase activity from other reported *Bacillus* species [20,27,95]. EDTA probably inactivated the cellulase activity either by removing metal ions from the enzyme through the formation of coordination complex or by binding inside the enzyme as a ligand [99]. The other organic solvents slightly inhibited enzymatic activity. Therefore, *B. mojavensis* strain REP303 was preferred for a broad range of applications in relation to organic solvents. The advantages of using enzymes in aqueous solutions containing organic solvents were the increased solubility of nonpolar substrates and the elimination of microbial contamination in the reaction mixture [100,101]. Today, almost all biotechnological applications require purified cellulases. The purification and polishing of enzymes are suggested for further study.

## 5. Conclusions

The isolation of cellulolytic bacteria from unique environments is a challenge for the acquisition of novel cellulases. Freshwater lakes provide a unique habitat for diverse bacteria because they differ from other aquatic habitats such as rivers and oceans. This study initially describes the diversity and cellulolytic activity of culturable aquatic bacteria from Tonle Sap Lake, the largest freshwater lake in Cambodia and Southeast Asia. There were three genera of cellulolytic bacteria isolated from sedimentary water samples of the lake belonging to *Aeromonas*, *Bacillus*, and *Exiguobacterium*. The results revealed that *B. megaterium* was the predominant species within the isolated cellulolytic bacteria. Moreover, the study also demonstrated that *B. mojavensis* strain REP303 was the most active cellulolytic bacterium. Finally, *B. mojavensis* strain REP303 and its endoglucanase could be considered as potential candidates for various biotechnological applications. Further studies are needed that concern the development of bacterial screening methods, the high throughput in silico analyses of enzyme structures, the genetic engineering of recombinant enzymes, and the purification of enzymes.

**Funding:** This research was funded by King Mongkut's University of Technology North Bangkok. Contract no. KMUTNB-64-DRIVE-33.

**Institutional Review Board Statement:** Not applicable.

**Informed Consent Statement:** Not applicable.

**Data Availability Statement:** The study did not report any data.

**Conflicts of Interest:** The author declares no conflict of interest.

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
