# Peer review of "Diversity and Activity of Aquatic Cellulolytic Bacteria Isolated from Sedimentary Water in the Littoral Zone of Tonle Sap Lake, Cambodia"

_water, doi:10.3390/w13131797_

Round 1
Reviewer 1 Report
This manuscript is quite well written apart from some English expressions, that need to be improved. The results are consistent with the objectives and I think it could be of interest for potential biotechnological application of the cellulases.
Title and Abstract, line 14: I suggest to remove the word cellulolytic before activity (it is redundant, since the authors already talk of cellulolytic bacteria), so only Diversity and activity of cellulolytic bacteria I think should be sufficient
The definition of "Sedimentary water" sounds quite strange, do you mean benthic boundary layer at the sediment–water interface, i.e. the layer between bed sediment and the overlying water column, or interstitial water extracted from the sediments? I suggest to change into benthic boundary layer or interstitial water.
Abstract, line 12, I would change " the prospects for application" into "in the perspective of their application"
In the Abstract please include a short description of the methodologies used in this study (as required by the Journal authors guidelines)
lines 14-18, avoid to repeat "aquatic" if you have already indicated the samples' nature (benthic boudary layer or interstitial water)
line 21, please indicate in full (carboxy methylcellulose) and after the abbreviation CMC) and explain that endoglucanases are a group of cellulases able to cleave the beta1-4 glycosidic bond present in CMC.
Introduction, line 36, please modify as follows. Microorganisms play a key rols in the decomposition of organic matter and nutrient regeneration, and are important members of microbial food webs as well as sensitive indicators of water quality [3-5]
You should add other references such as Song C et al. (2019) Nutrient regeneration mediated by extracellular enzymes in water column and interstitial water through a microcosm experiment, Science of The Total Environment, 670, 982-992,https://doi.org/10.1016/j.scitotenv.2019.03.297.
Caruso et al. (2010) Assessment of the ecological status of transitional waters in Sicily (Italy): First characterisation and classification according to a multiparametric approach. Marine Pollution Bulletin, 60, 1682-1690, https://doi.org/10.1016/j.marpolbul.2010.06.047.
Zhang H, Sekar R and Visser PM (2020) Editorial: Microbial Ecology in Reservoirs and Lakes. Front. Microbiol. 11:1348. doi: 10.3389/fmicb.2020.0134
line 53, organic matter
line 55, instead of "are comprised of"use "comprise" or "include"
Materials and methods
line 105, modify to obtain with to culture aerobic heterotrophic bacteria
line 108, instead of The samples were taken for ....change with The collected samples were stored at + 5°C before their analysis, performed within 48 hours from sampling
line 117, of the growth medium
line 119, at a concentration
line 124, purified by streaking on plates of a selective medium
line 126, Agar, previously described by...., contained 0.2 g
line 133 molecular methods (delete genetic)
Minor revision of English language is suggested, i.e. on line 136, of each isolated strain (not bacteria), on line 168, instead of The iodine solution comprised, please change into The iodine solution contained; lines 170-171, The cellulolytic bacteria able to produce....on CMC Agar through their extracellular cellulases.
line 184, the concentrated supernatants served as crude cellulase extracts, and kept ....
lines 275 and 286, Crude cellulase extract
line 282, Instead of They use "All the reagents" or "Cellulase extracts"
Results
line 301, remove by investigations, were of suspended sediments, and light-brow....
line 302, remove with instrumentation (already detailed in Mat. and methods)
line 308, Heterotrophic bacteria (n=83)
line 311, However, only 14 bacterial isolates could be cultivated
line 313 strains were considered as axenic isolates of cellulolytic bacteria.
line 346, The bacterial isolates
lines 356 and 358, bacterial isolate
line 361, The cellulolytic bacteria were found to be closely related
line 374, instead of cellulolytic performance use enzymatic activity
line 397, crude cellulases having from... to
line 407, analysis of endoglucanase was performed on..
Paragraph 3.5 please move lines 470-473 before lines 468-470, and lines 479-482 before line 476
Discussion
line 527, remove sampling water(only samples)
line 529, is composed by (instead of is comprised of)
line 542, highly efficient cellulolytic bacteria
line 546, Molecular identification (instead of Genetic)
line 553, endospores (add a reference)
line 568, was the most abundant cellulolytic species, accounting for 42%
line 574, defined or characterized?
lines 580 and 586, was an active cellulolytic bacterium
line 584, have been confirmed to be cellulolytic bacteria
line 594, according to (instead of supported by)
line 600 A. jandaei has been isolated...and has been associated....
Move lines 604-605 close to the sentence on line 602, talking about bacterial diseases in humans.
lines 617-618, were the main responsible for cellulolytic activity
line 643, Crude cellulase extracts
Table 1 caption: samples (remove "es" at the end)
Table 2 caption: Identity percentages of 16S.... of the 14 cellulolytic bacteria with closely-related bacteria
Table 3 caption: HC values and enzyme activity of....
Conclusions
The results revealed that B. megaterium was the species predominant within the isolated cellulolytic bacteria. Meanwhile, they evidenced that also B. mojavensis strain REP303 was the most active cellulolytic bacterium. Finally, B. mojavensis strain REP303 and its endoglucanase could be considered as potential candidate for various biotechnological applications.
Author Response
Dear Reviewer 1, I would like to thank for your nice suggestions. Please see the attachment. Sincerely.
Reviewer 2 Report
This is a well conceived and executed study investigating the diversity of carbohydrate degrading bacteria in a previously under-explored lake ecosystem. The study is a nice mix of cultivation, bioinformatic, and biochemical work. I only have a few small comments for the author to consider.
Line 19. What's this sentence mean? It’s poorly worded.
Line 21. what does CMCase mean? Please define.
Lines 36-41. Cyanobacteria have a very different role that OM decomposition. Please rewrite more broadly on the role of bacteria, or focus on the heterotrophs.
Line 53. no ‘s’ on matters.
Lines 63-65. References for this statement?
Line 70. Awkward sentence. “has been conducted” or “ should be conducted”?
Lines 74-76. Is there reason to believe the bacteria and cellulases will be different than in other lakes?
Figure 1. Can you add borders instead of stars to show the countries?
Line 113. why use saline water for sample collection and preservation?
Line 130-131. CMC is not a sole carbon source, there's peptone and agar (another polysaccharide) in the medium that can fuel heterotrophic growth.
Line 299. is "randomly" really necessary in this statement? I doubt it was truly random sampling. And wouldn’t nonrandom sampling of different regions make more sense to capture the bacterial diversity?
Line 300. I suggest changing “morphological” to “environmental” here and elsewhere.
Line 310. Still not true, there’s other organic carbon compounds in the medium.
Line 326. Change to “the phylum”
Line 416. Can you report the GH family here?
Line 551. Remove ‘to’
Line 629: How do you know this is the active enzyme? There could be other GHs operating in the cell? How might your story change if you sequenced the whole genome? Or did proteomic work on the supernatant?
Author Response
Dear Reviewer 2,
I would like to thank for your nice suggestions.
Please see the attachment.
Sincerely.
